# The *MUC2* Gene Product: Polymerisation and Post-Secretory Organisation—Current Models

**DOI:** 10.3390/polym16121663

**Published:** 2024-06-12

**Authors:** Kyle J. Stanforth, Maria I. Zakhour, Peter I. Chater, Matthew D. Wilcox, Beth Adamson, Niamh A. Robson, Jeffrey P. Pearson

**Affiliations:** 1Aelius Biotech, The Medical School, Framlington Place, Newcastle-upon-Tyne NE2 4HH, UK; peter.chater@aeliusbiotech.co.uk (P.I.C.); matthew.wilcox@aeliusbiotech.co.uk (M.D.W.); beth.adamson@aeliusbiotech.co.uk (B.A.); niamh.robson@aeliusbiotech.co.uk (N.A.R.); 2Biosciences Institute, Newcastle University Biosciences Institute, Catherine Cookson Building, Framlington Place, Newcastle-upon-Tyne NE2 4HH, UK; m.zakhour1@newcastle.ac.uk (M.I.Z.); jeffrey.pearson@newcastle.ac.uk (J.P.P.)

**Keywords:** MUC2, mucus barrier, mucins, polymerization, cystic fibrosis, gastrointestinal tract

## Abstract

MUC2 mucin, the primary gel-forming component of intestinal mucus, is well researched and a model of polymerisation and post-secretory organisation has been published previously. Recently, several significant developments have been made which either introduce new ideas or challenge previous theories. New ideas include an overhaul of the MUC2 C-terminal globular structure which is proposed to harbour several previously unobserved domains, and include a site for an extra intermolecular disulphide bridge dimer between the cysteine 4379 of adjacent MUC2 C-termini. MUC2 polymers are also now thought to be secreted attached to the epithelial surface of goblet cells in the small intestine and removed following secretion via a metalloprotease meprin β-mediated cleavage of the von Willebrand D2 domain of the N-terminus. It remains unclear whether MUC2 forms intermolecular dimers, trimers, or both, at the N-termini during polymerisation, with several articles supporting either trimer or dimer formation. The presence of a firm inner mucus layer in the small intestine is similarly unclear. Considering this recent research, this review proposes an update to the previous model of MUC2 polymerisation and secretion, considers conflicting theories and data, and highlights the importance of this research to the understanding of MUC2 mucus layers in health and disease.

## 1. Introduction

From the mouth to the anus, the surface area of the mucosal epithelium of the digestive tract covers approximately 32 m^2^ [1]. These specialised epithelial layers are continually exposed to aggressive and potentially damaging physical, chemical and biological conditions, including physical damage and shear forces from solid and potentially abrasive foods, gastric acid, bile, digestive enzymes and both symbiotic and pathogenic microorganisms [2,3,4]. The mucus barrier plays an essential role here, coating these epithelial surfaces (with the exception of the oesophagus and the mouth [5,6,7] and protecting the delicate tissues from these challenges while allowing absorption, secretion and host-microbe cross talk [8]. Outside of the digestive tract, mucus has a similar function for other epithelial surfaces such as the airways, the eyes, and the cervical-vaginal tract [7].

Mucus is a complex functional biopolymer with variable properties of a viscous liquid, or an elastic solid, depending on type, site and physiological conditions [9]. Epithelial surface mucus forms heterogeneous hydrogels, with water occupying up to 95% of its total volume. The remaining 5% is constituted by DNA, lipids and proteins, much of which is debris produced during epithelial cell shedding [7,8], and mucin, a highly O-glycosylated glycoprotein which is the primary gel-forming component of mucus gels [7]. Mucins can be categorised into either two subgroups, membrane-tethered or secreted [10,11,12,13], or three subgroups, membrane-tethered, secreted gel-forming and secreted non-gel-forming [7]. Secreted gel-forming mucins are those which contribute to the formation of epithelial surface mucus, being the major gel-forming component of these mucus layers.

Epithelial surface mucus layers are achieved through the formation and hydration of extensive mucin biopolymers linked via intermolecular disulphide bridges. Disulphide bridge-mediated polymerisation occurs through domains also found in von Willebrand factor, another important functional biopolymer which supports haemostasis [6]. These biopolymers hydrate significantly on secretion and form the mucus gel, and without polymerisation no gel formation can occur. This makes secreted mucins very important biopolymers which are crucial for the function of many epithelial surfaces throughout the human body.

Of these mucins, the MUC2 mucin, which forms the mucus layers of the small intestine and colon, has often been considered as the most researched and understood secreted gel-forming mucin, with several models of polymerisation and secretion being published in the literature [6,7,14,15,16,17,18,19,20]. While recent publications continue to add to and update these models [20], other publications call these models into question by giving experimental evidence which is contrary to that of the previous models [6,19]. In this review we highlight the structure and function of mucins in the human body including each different category of mucin, and later review the literature surrounding the structure, polymerisation and post-secretory organisation of MUC2 mucins/mucin polymers.

## 2. Mucin Structure

Mucins are large molecular weight, heavily O-glycosylated glycoproteins. Mucins, within their categories, share several common domains and regions which help identify the molecules as mucins while also maintaining a level of structural diversity and functional uniqueness. This is achieved via specific variations in the type of domains present, molecule length and presence of extra non-shared domains unique to specific mucins. Membrane-tethered, secreted gel-forming and secreted soluble mucins differ significantly in their structure. Secreted gel-forming mucins (Table 1; Figure 1) typically share a number of domains important for polymerisation, specifically von Willebrand factor-like D1, D2, D’ and D3 domains and a cysteine knot domain, which together are found in all gel-forming mucins and promote gel formation [21]. Membrane-tethered and secreted non-gel-forming mucins do not contain these important polymerising domains, and thus do not form gels. Membrane-tethered mucins instead harbour a transmembrane region which anchors them to the cell surface, allowing the mucin to contribute to the glycocalyx [22]. Secreted non-gel-forming mucins are typically unique in structure and share little in common with other mucins, even those within the non-gel-forming group [23,24,25,26].

Despite their differences, all mucins contain a key structure which makes the molecules easily distinguishable from other non-mucin glycoproteins. This region is the site of the heavy O-glycosylation associated with mucins, and is known as the variable number tandem repeat (VNTR) region [7,11,13,27]. Often occupying around half or more of the mucin protein core, the VNTR region contains a characteristic, variable but repetitive series of amino acid residues consisting of serine, threonine and proline (STP), which varies internally between each individual mucin type. For example, there are one hundred sixty-nine amino acids present in the repeat sequence for MUC6 [28], compared to the twenty-three amino acids for MUC2 [28], twenty-nine for MUC5B [29] and eight for MUC5AC [30]. The VNTR region is critical in the function of mucin, as it is saturated with sites for O-glycosylation, allowing for the generation of a vast network of carbohydrate side chains that can occupy 70% or more of the total mass of the glycoprotein [31], leading to molecular weight ranges of 0.2–10 MDa, dependent on VNTR length and mucin type [7,11]. This feature allows membrane-tethered mucins to contribute to the formation of the glycocalyx, a vast network of sugars, presented on the surface of cells consisting of transmembrane glycolipids and glycoproteins, and protect delicate epithelial cells from invasive, pathogenic microorganisms [32]. The presence of carbohydrate chains on secreted mucins gives them the ability to form a gel where the carbohydrate side chains readily associate with water molecules, allowing for significant hydration [33].

O-glycosylation is the covalent linkage of N-acetyl galactosamine (GalNAc) by peptidylgalactosaminyltransferases (GalNAc-Ts) [34]. Up to 20 GalNAc-Ts have been identified in humans [35], and expression can vary based on tissue type. For example, Arike et al. (2017) found that some GalNAc-Ts were more abundant in the colon in comparison to the small intestine. Following GalNAc linkage, a variety of glycosyltransferases begin a progressive elongation of the carbohydrate chain utilising four specific sugar groups: fucose, N-acetyl glucosamine, galactose, and sialic acid, which is a type of N-substituted neuraminic acid. Chains developed by this process can reach up to 20 residues in length [7,13,27]. There are many glycosyltransferase enzymes, and their expression can vary between organs and tissues [34], allowing glycoproteins, including mucins, to have variable glycosylation patterns. Despite not being a direct site of O-glycosylation, proline has been identified as a facilitator of carbohydrate linkage to serine and threonine residues [36], particularly when the proline residue is three amino acids in front (+3) or one behind (−1) [37]. This is thought to explain its abundance and proximity to serine and threonine in the VNTR region [36]. Sugar groups present in the carbohydrate chain can be subject to sulphation—a process thought to increase resistance to pathogenic microorganisms by reducing penetrability in colonic mucus [38]. Data from previous studies have shown that reduced MUC2 sulphation has been observed in mucins present in ulcerative colitis patients [39]. Glycan sulfation, combined with terminal sialic acid residues, contributes significantly to the overall negative charge of mucins like MUC2 at physiological pH [7]. Negative charges in mucus have been associated with stiffer mucus gels [40].
Figure 1Diagrammatic representation of secreted gel-forming mucins. Legend is presented below the diagrams indicating each specific domain. vWF-like D domains are well conserved between MUC2, MUC5AC and MUC2. MUC6 lacks a D4 domain, and MUC19 lacks a D4 and D’ domain. Conserved throughout all of these mucins is a CK (cysteine knot) domain. B and C domains are well conserved between MUC5AC and MUC5B, with MUC19 and MUC2 lacking a B domain and MUC6 lacking both. MUC2 has four C domains between the D4 domain and CK domain in the C-terminal regions, which are in series: C1, C’, C3 and C4. CYS domains are present in MUC2, MUC5B and MUC5AC. Non-repetitive domains are found in MUC5B, MUC5AC, MUC6 and MUC7. MUC6 has an incredibly long VNTR region, while MUC19 has an unusually long N-terminal sequence. Images are not to scale. References taken from [8,20,41,42].
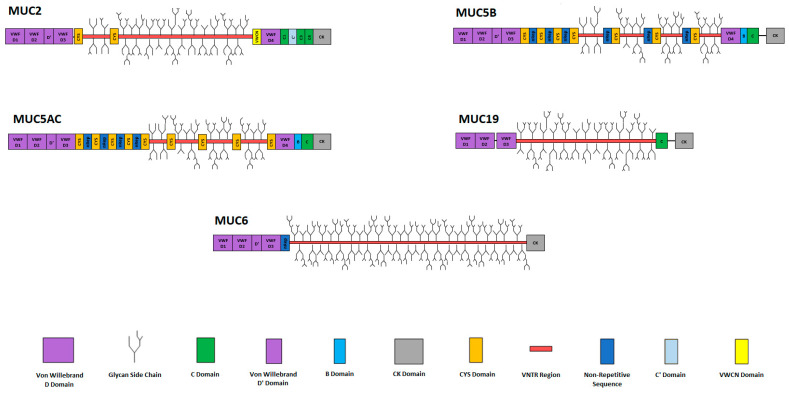



## 3. Secreted Mucins

There are two types of secreted mucin. MUC7, MUC8 and MUC9 mucins are non-gel-forming secreted mucins, whereas MUC2, MUC5AC, MUC5B, MUC6 and MUC19 mucins are all secreted gel-forming mucins which form the epithelial surface mucus layers which protect epithelial surfaces throughout the body. The presence of each mucin varies between tissues. For example, MUC2 mucins are the primary gel-forming mucin of the small intestine and colon [15,16,18,34,43,44,45], whereas a combination of MUC5B and MUC5AC mucins constitute the airway mucus layer [46,47,48,49], and the stomach is coated with a mucus layer formed by MUC5AC and MUC6 mucins [41].

### 3.1. Secreted Gel-Forming Mucins

The general structure of gel-forming, secreted mucins is detailed as follows. Individual mucin structures can be seen in Figure 1. At the N and/or C termini of secreted mucins there are von Willebrand factor-like (vWF) domains, which resemble the globular D domains of the von Willebrand factor (Figure 1) [50]. A typical vWF D domain is constituted by a vWD module, C8 (a domain containing eight conserved cysteine residues), a trypsin inhibitor-like module (TIL) and an E module. The D’ domain contains only a C8 and TIL sequence [51]. Some D domains, like the D4 domain of vWF [51] and MUC2 do not contain an E module [20]. The N-terminal domain organisation of secreted gel-forming mucins house three of four vWF-like D domains and a half D domain; in order, these are D1, D2, D’ and D3. MUC19 is the only secreted gel-forming mucin proposed to lack a D’ domain [8]. At the C-termini, a number of other vWF-like domains can be found; these include the fourth full D domain (D4), and a combination of von Willebrand C, von Willebrand B and cysteine knot (CK) domains. Of these C-termini domains, the CK domain is highly conserved throughout all secreted gel-forming mucins [7,8,52,53,54]. The presence of the other domains varies between each mucin. The D4 domain is found at the C-terminus of MUC2, MUC5AC and MUC5B mucins [7,8], and B domains are found in the C-terminus of MUC5AC and MUC5B mucins [7,8]. It was previously thought that a B domain was found within the C-terminus of MUC2, with the C-terminus structure consisting of, in order, D4-B-C-CK (Figure 1) [7]. However, Gallego et al. (2023) [20] used cryo-electron microscopy and AlphaFold modelling to study the C-terminal structure of MUC2, and have suggested that a B domain is not present, and that the C-terminus is instead constituted, in order, by the following: VWCN (a globular structure with a vWC fold), D4, C8, a trypsin inhibitor-like domain (TIL), three von Willebrand C domains with a C’ (half C domain) between C1 and C3, and a final highly conserved cysteine knot. C domains are also present in the C-terminus of MUC5AC, MUC5B [52] and MUC19 [53].

These vWF-like domains are globular structures that are important in the polymerisation of secreted mucins [7,8,45,55]. These domains contain up to 10 half-cystine residues, with cystine being the oxidised dimer form of cysteine—these residues occur when the R group of cysteine, a thiol group (SH), is deprotonated to give a thiolate (S^−^) [56]. These residues contribute to polymer organisation through the formation of inter-molecular disulphide bridges as well as non-covalent interactions [7,55]. Non-covalent interactions are the result of the hydrogen bonding of the nearby positively charged amino acids histidine, lysine and arginine [56]. Covalent and non-covalent interactions serve a variety of functions at different stages of mucin synthesis and assembly e.g., the dimerization of MUC5AC, MUC5B and MUC2 is known to occur at the CK domain in the endoplasmic reticulum via the formation of disulphide bridges, before further polymerisation in the Golgi at the D3 domain, again via disulphide bridge formation [8,27,52]. Moreover, positively charged histidine residues non-covalently interact with the thiolate groups of deprotonated cysteine residues which allow polymerisation and packaging into granules for secretion [7,15,43,52].

The central region of secreted mucins, between the vWF domains of the N and C termini, contain the VNTR region, sometimes described as the mucin fingerprint region, which is the site of extensive O-glycosylation. The glycan side chains, which extend out from the central protein core of the VNTR region of mucins, are hydrophilic, meaning that under normal circumstances they would dissolve in water. Mucus however does not dissolve in water due to the formation of non-covalent interactions between mucin polymers. The hydrophilic glycan side chains contain many OH groups, which due to the unequal sharing of electrons result in electronegative oxygen and electropositive hydrogen atoms, giving them the ability to form hydrogen bonds with water molecules. This allows the mucins to take on significant amounts of water and expand 1000–30,000-fold on secretion. This is facilitated by an exchange of Ca^2+^ ions, which cross-link negative charges of mucin polymers keeping them tightly packed until they are secreted, for Na^+^ ions, which results in an uncoupling of Ca^2+^ cross-links allowing mucins to expand [7]. The negative charges present within the VNTR glycan chains are due to the presence of sialic acid and ester sulphation, and while counterions such as Na^+^ may interact and neutralise these charges, many will still be exposed and give the mucus the ability to trap cationic particles and repel anionic particles [7,55]. Like cell surface mucins, the VNTR-associated glycans are protective as they act as ligands for pathogenic bacteria, but they are also a good source of carbohydrate for commensal bacteria, and play a large role in the regulation of the microbiome [12].

Although not conserved throughout all secreted gel-forming mucins, CYS domains are hydrophobic cysteine rich regions found in the central region of MUC5AC (4), MUC5B (6) [27,52] and MUC2 (2) [7,45]. The presence of these domains is strongly associated with the presence of vWF-like domains within the same protein backbone in mammals, and is very well conserved throughout a number of taxonomic classes, with 10 consistent cysteine residues found within the CYS domain, from sea urchins to humans [57]. These CYS domains, similarly to vWF-like domains, are saturated with half-cystine residues which form intramolecular disulphide bridges, contributing to the tertiary structure of the mucins. Moreover, hydrophobic interactions are thought to contribute to the formation of mucin–mucin CYS dimers, which in turn are suggested to modulate the pore size of the mucus by creating stiffer gels [15].

### 3.2. Secreted Non-Gel-Forming Mucins

Three secreted soluble mucins have been identified, MUC7, MUC8 and MUC9. These are a diverse group of non-gel-forming mucins which share little structural and functional homology and are identified as mucins through the presence of a mucin type VNTR. These mucins do not form gels as they lack the ability to polymerise due to the absence of polymerisation domains observed in other mucins (e.g., vWF-like D domains and cysteine knot domains at the N and C termini) [8].

MUC7 is a unique mucin expressed in the sublingual gland and the submandibular glands of the oral cavity, making it a salivary mucin. The gene which codes for this mucin can be found at chromosome locus 4q13-21 and is 150 kDa in size [58]. The MUC7 glycoprotein is 357 amino acids in length [59] and has a much simpler structure than the other secreted mucins. At the N-terminus resides a histatin-like domain which is free of glycosylation. This is followed by a short, O-glycosylated VNTR region composed of five or six, 23 amino acid-long STP repeats, flanked by two non-repetitive, O-glycosylated and N-glycosylated sequences [8]. A final structural motif, a leucine zipper, is found at the C-terminus [59].

In terms of secretion, there is very little information on MUC7. The structural components, however, and lack thereof, do give insight into the functionality of the mucin. The histatin-like domain is aptly named as it has both structural and functional resemblance to the antimicrobial salivary histatin Hsn-5, where it has been observed to exhibit potent bactericidal and fungicidal properties against the pathogenic organisms *Candida albicans, Cryptococcus neoformans, Streptococcus mutans* and *Streptococcus gordonii* [60]. It may also be speculated that VNTR-associated carbohydrates may also contribute to the agglutination of harmful microbes, as MUC7 has been shown to have binding capacity for *S. pneumoniae* and *S. sanguinis*, the latter exhibiting binding specific to terminal sialic acids [23]. The lack of vWF-like domains means that MUC7 will not organise into complex polymers like the gel-forming mucins. The C-terminal leucine zipper has been suggested to function in the dimerization and secondary structure stabilisation of the MUC7 mucins [61]. It is thought to stabilise these mucins, but also allow the formation of dimers. Leucine zippers are identifiable by the periodic repeats of leucine residues at every seventh position. This results in the formation of an alpha-helix structure. Helix–helix interactions between these leucine-based helical structures promote dimerization which is stabilised by leucine side chains which form an inner hydrophobic core [62]. The formation of MUC7 dimers via the leucine zippers may enhance its antimicrobial properties, possibly by promoting agglutination [61].

MUC8 is expressed in the airways and located at gene locus 12q24.3 [7]; the full length cDNA sequence of MUC8 has not been determined, but a VNTR has been identified [25]. MUC8 function is not well understood, but it has been found that MUC8 is upregulated in the airways of patients with chronic rhinosinusitis [25,63], cystic fibrosis [64] and chronic airway inflammation [65]. It is thought that MUC8 may induce a reduction in the ATP/P2Y_2_-mediated activation of IL-1 and IL-6 in inflammatory responses, regulating the airway inflammatory response by acting as an anti-inflammatory [25].

MUC9 is found at gene locus 1p13.2 and is expressed in the oviduct [7], and referred to as Oviduct-Specific Glycoprotein 1 (OVGP1) [26]. Studies have suggested that MUC9 in humans is important for sperm/oocyte binding and zona pellucida penetration [66]. Algarra et al. (2016) suggest that the C-terminal region forms an association with the zona pellucida in such a way that increases penetrability to sperm cells. MUC9 is classified as a mucin through the presence of a VNTR region, and shares significant sequence homology at the N-terminal region with the glycoside hydrolase 18 family; however, the region is enzymatically inactive due to the lack of an essential glutamic acid residue [26]. The C-terminal region of MUC9 is highly variable between species, and four regions have been identified; these are noted as B, C, D and E, and the exact functions of these regions are yet to be fully understood.

## 4. MUC2: The Major Gel-Forming Mucin of the Small Intestine and Colon

The MUC2 mucin, of the gene locus 11p15.5 [7], is considered one of the most studied and well understood mucins within the current literature landscape, and was one of the first to have its polymerisation, secretion and post-secretory organisation detailed in a way that allowed the development of a theoretical model [7,8,14,15,16,43,44]. Research on MUC2 mucins also allowed for a more thorough understanding of gel-forming mucins as a whole. MUC2 mucins are known for their role in the formation of the epithelial surface mucus gel of the small intestine [7,67,68,69] and the colon [44,45,69], where they are the major gel-forming component of the secreted mucus which protects the epithelial surface of these tissues.

### Structure and Polymerisation of MUC2 Mucins

The structure of MUC2 (Figure 1) was briefly highlighted in the previous section. At the N-terminal side of the mucin monomer, four vWF-like D domains are found in series in the order of D1, D2, D’ and D3, with D’ representing a half D domain. The central region begins at the end of the D3 domain. Here, a short VNTR region, flanked on either side by a CYS domain, precedes a significantly longer VNTR region which spans the majority of the backbone [7,8,45]. The shorter of the two VNTR regions is composed of a highly conserved STP repeat region of around 26 amino acid residues [8]. Based on individual allele analysis, the second, lengthy VNTR region contains STP repeats consisting of 23 residues in length, of which the number of repeats can vary from 40 to around 185 [8,70]. Therefore, the second VNTR sequence can vary from around 920 residues to 4255 residues in length, contributing to a mucin backbone of around 5000 residues in length. This will cause significant variation in the number of glycan chains found between individual MUC2 mucins, due to a large difference in the number of glycan target serine or threonine residues, and may have importance to specific diseases, as short alleles in other gel-forming mucins have been found to confer susceptibility to pathogens such as *Helicobacter pylori* [71]. These lengthy VNTR regions, accompanied by glycan chains, contribute to the mucins’ large molecular mass of around 1.5 million Daltons [72]. Following the VNTR region, the C-terminal side of the backbone was thought to be composed of, in order, D4, B, C and CK [50]. However, as highlighted previously, Gallego et al. (2023) [20] used cryo-electron microscopy and Alphafold modelling to study the MUC2 C-terminal structure and proposed an updated C-terminal structure of, in order, VWCN-D4-C8-TIL-C1-C’-C3-C4-CK [20]. This is the C-terminal structure shown in Figure 1. The research group also suggests the presence of novel, previously unknown intermolecular interactions between the C-terminal structures of separate MUC2 mucins, namely the formation of an intermolecular dimer at the N-terminal side of the C-terminal structures through the presence of a disulphide bridge between two VWCN domains, and interactions between the C1 domain and the C-terminal structures of adjacent MUC2 mucins. The details of these interactions will be discussed further as we discuss the literature surrounding the polymerisation and post-secretory organisation of MUC2 mucins.

Of the secreted gel-forming mucins, the polymerisation of MUC2 mucins has been studied in the most detail and is arguably the most well understood. A consensus model has been presented in the literature previously, and recent publications continue to add further detail to this model. However, while many areas of this model are accepted, there are other parts of the model where there are conflicts in the literature.

The first step of the dominant model of MUC2 monomer polymerisation and secretion [7,15,18,43,45] begins immediately after mRNA translation in the endoplasmic reticulum. Here, at pH 7.2, the MUC2 monomers dimerise at the C-terminal CK domain. These dimers are then translocated to the Golgi apparatus to undergo a further trimerization at vWF-like D3 domains. This is achieved through the formation of disulphide bridges between deprotonated thiol groups present on cysteine residues within each individual domain. The bulk of MUC2 VNTR O-glycosylation also takes place at this stage, predominantly through core 1–3 structures [73]. Glycan cores one and three are linear, having either a galactose (core one) or N-acetyl glucosamine (core three) attached to the serine/threonine-associated GalNAc residue prior to chain elongation. Glycan core two is a dual branched core containing both galactose and N-acetyl glucosamine, respectively [74]. A pH drop to 5.2 within the Golgi apparatus causes nitrogen atoms associated with histidine residues to readily accept protons, giving it a positive charge and promoting non-covalent linkages between the D1, D2 and D3 domains. This allows mucins to form 5/6 sided lattice-like structures, which contain D3 domains at the corners. An increase in free calcium ions, a divalent cation, within the Golgi allows the cross-linking of negative charges from ester sulphates and sialic acid residues in the glycan chains, promoting the tight packaging of mucins into secretory vesicles. Following the secretion of the mucin granules, free sodium ions displace the calcium from within the mucin granules, which are then chelated by luminal bicarbonate. This ion exchange releases the cross-links, allowing the mucins to expand, revealing the extensive glycan network. The glycan network then readily associates with water molecules, allowing the mucus to hydrate significantly and swell up to 3000-fold. Within the newly formed mucus layer, the lattice-like structures form pores of roughly 200 nm in diameter [75].

There are several studies which contrast the way that this model describes MUC2 polymerisation. The first of these is a recent study by Gallego et al. (2023) [20], which as previously discussed, has used experimental evidence to propose an update of the MUC2 C-terminal globular domains from D4-B-C-CK to VWCN-D4-C8-TIL-C1-C’-C3-C4-CK. A 774 amino acid long MUC2 C-terminus (amino acid sequence taken from the MUC2 amino acid sequence published by Svensson et al. (2018) [28]), was expressed in Chinese hamster ovary cells. During the study, which utilised cryo-electron microscopy and Alphafold modelling, the MUC2C was found to form disulphide bridges at both ends. Three disulphide bonds were known to form at the CK domain which promotes dimerization of MUC2 monomers in the endoplasmic reticulum, but this is the first noted instance of a disulphide bridge located at the N-terminal side of the C-terminal region. Further analysis revealed that this C-terminal region is comprised initially of a compact globular region of a VWC fold and a D4 domain, followed by a stalk of VWC domains (C1-C’-C3-C4) and a CK forming the tail. The C-domains form a hinge-like structure allowing the C domains and the C-terminal regions to be flexible. Gallego et al. [20] observed two intermolecular interactions which contribute to C-terminal dimers. A dimer is formed at the VWCN domain through hydrophobic interactions between isoleucine 4394, valine 4392, phenylalanine 4382 and leucine 4378, and an antiparallel beta sheet interaction between threonine 4391 to glutamate 4393. The noted extra disulphide bridge is formed between the cysteine 4379 of adjacent MUC2 C-termini. The researchers note that while the other interactions are present in other secreted mucins like MUC5B and vWF, the disulphide bridge is not present due to the lack of cysteine 4379. The C1 dimer forms via two interfaces: the first is the formation of hydrogen bonds between glycine 4795 and glutamic acid 4793 and the D4 domain of an adjacent MUC2 C-terminus, and the second is the formation of hydrophobic interactions between the C1 domains of adjacent MUC2 mucins mediated by proline 4771, phenylalanine 4784, and isoleucine 4798. The elucidated structure of the MUC2 C-terminal region here is a significant deviation from the previously assumed C-terminal structure. There is an argument for the incorporation of this C-terminal structure into the existing model of MUC2 polymerisation and secretion previously discussed, particularly considering the discovery of disulphide bridge formation at either end of the C-terminal region, and how this may influence the polymerisation and function of MUC2 mucus gels. Of course, this study was performed using only the MUC2 C-terminus which lacked the N-terminus and VNTR region which occupy the vast majority of the protein backbone. Therefore, it is unclear whether this extra disulphide bridge forms when whole mucin monomers are involved. Research on whole mucins is notoriously difficult, however, and the basis of much of the model is also based on the expression of truncated MUC2 mucins which lack the vast majority of the mucin protein backbone. For example, Godl et al. (2002) [76] expressed a truncated MUC2 protein which contained only the N-terminus. Electron microscopy was used to analyse the organisation of the MUC2 N-terminal, finding that they formed 240 kDa structures, which upon reduction formed three 85 kDa structures, this being the first evidence that the MUC2 N-terminus forms a trimer. Thus, introducing the findings by Gallego et al. (2023) [20] into the model is reasonable.

Furthermore, contrary to the previously discussed model, and casting doubt over the accuracy of the model, Javitt et al. (2019) [19] have provided experimental evidence which suggests that MUC2 undergoes a formation of D3 dimers, not trimers [19], contrasting with the work of Godl et al. (2002) [76] and Nilsson et al. (2014) [18] whom both suggest D3 trimer formation based on their own experimental evidence. As previously, the 2002 study by Godl et al. [76] forms part of the backbone for the theory of trimer formation in MUC2 D3 domains. Over a decade later, Nilssen et al. [18] contributed further to the trimeric model of MUC2 polymerisation. In these experiments, CHO cells were used to express three different recombinant short MUC2 fragments containing the D’D3 domains alone (MUC2D3), and D’D3 domains, a CysD1 domain and a Myc tag, one without GFP (MUC2D3CysD1-M) and one with GFP (MUC2D3CysD1-MG). Through gel filtration, electron microscopy and single particle processing, their results suggested that, not only do the domains form trimers, but the trimers also interact with one another non-covalently to form six domain globules which contribute to the 3D structure and functionality of MUC2 as intestinal mucus. Javitt et al. (2019) [19], however, achieved results that were contrary to those found in the previous two sources. Using the human embryonic kidney (HEK) -293F cellular expression system and a similar MUC2D3 recombinant MUC2 fragment to Nilssen et al., they found that the MUC2D3 protein has a mass of 115 kDa in its oligomeric form under non-reducing conditions. This is much smaller than what was found previously (around 240 kDa in size), but a significant effort to reduce the N-glycosylation of the mucin fragment was made, which could explain some of the variation, along with the fact it is not an identical fragment. However, under reducing conditions the fragments separated into two smaller fragments between 55 and 70 kDa in size, indicating the presence of a dimer. They also state that the disulphide bridges form between the C^1088^ECFC and C^1130^EWH residues. Javitt et al. [19] go on to suggest that MUC2 may likely dimerise similarly to the vWF, MUC5AC and MUC5B, as opposed to forming trimers. In 2020, a second study was published by Javitt et al. [6] in which they continued their work, using new analytical techniques to investigate MUC2 polymerisation with a particular focus on the D3 domain. HEK293F cells were used to express a truncated MUC2 protein, referred to as the “head” of the protein, which consisted of the N-terminal globular structures D1, D2, D’, D3 and CysD. This MUC2 “head” was imaged and analysed using cryo-electron microscopy. At pH 5.4–6.2, the MUC2 head formed a linear filamentous structure composed of beads. Cryo-electron microscopy on vitrified MUC2 heads found that the beads were formed by interacting MUC2 heads. Each MUC2 head spanned across two beads, with one bead containing the D1 and D2 domains of one MUC2 head, and a D3 domain from an MUC2 head where its respective D1 and D2 domains constitute the adjacent bead. The D3 domain of the first MUC2 head is similarly donated to the adjacent bead. Javitt et al. [6] suggest that a TIL1/E1 region of the D1 domain acts as a cradle for the D3 domain donated by the adjacent bead, with the TIL2/E2 of the D2 domain, and the TIl’/E’ of the D’ domain, forming a bridge between the beads. Within these beads, the D3 domain was found linked to a juxtaposed D3 domain of an adjacent MUC2 head pair, forming a dimer. Javitt et al. [6] suggest that the non-covalent interaction is essential to the formation of these D3 dimers. Observing similarities between the filamentous structure of the MUC2 heads and the vWF tubules that form prior to vWF secretion, Javitt et al. [6] proposed a new model of MUC2 polymerisation prior to secretion. Firstly, the MUC2 monomers form a disulphide-mediated dimer at the CK domain. Following this, the N-terminal domains organise into a bead assembly where the D3 of one monomer is associated with the TIL1/E1 cradle of the other. The dimer then undergoes filament formation with other MUC2 dimers where juxtaposed D3 domains form intermolecular disulphide bridges, and upon secretion the non-covalent interactions within the cradle release, forming a disulphide-linked linear polymer like that observed for MUC5AC and MUC5B.

Overall, where there was once a consensus model of MUC2 mucin polymerisation, recent work has led to the polarisation of this field. The absence of an MUC2 trimer is significant when considering the organisation of MUC2 mucin polymers. Trimer formation theory generally assumes that the trimers form the hinges of five or six-sided ring-like structures that non-covalently interact with other mucin polymers, giving a gel-forming polymer network. A lack of trimer formation, as suggested by Javitt et al. (2019; 2020) [6,19] assumes that MUC2 mucins form linear polymers similar to that of MUC5B and MUC5AC mucins. What is clear is that much more research is required in this field to confidently identify how MUC2 polymerises. Given the difficulty in studying whole mucin monomers and polymers, this may rely on the development of new technologies.

## 5. Small Intestinal Mucus Layer Post-Secretory Organisation

Like polymerisation, the organisation and function of secreted MUC2 mucus layers present in the small intestine is similarly not well characterised, with varying opinions in the literature surrounding the formation of an inner and outer mucus layer—a phenomenon observed in the colon from the *MUC2* gene product [44]. Over the last decade however, there has been a substantial increase in research papers focusing on the secretion and post-secretory organisation of MUC2 polymers in the small intestine. Many of these ideas stem from the point of a dysfunctional chloride/bicarbonate channel (Cystic Fibrosis Transmembrane conductance Regulator: CFTR) contributing to the cystic fibrosis phenotype in the small intestine [17], and may allow for a deeper understanding of how the mucus layer in the small intestine contributes to the overall function of the organ, thus satisfying the parameters for small intestinal function stated in the previous section. In this section we will review evidence for the post-secretory processing of MUC2 polymers in the small intestine, as well as an evidence-based analysis of the presence of, or lack of, an inner layer of mucus in the small intestine. 

### 5.1. CFTR, Bicarbonate and Meprin β Mediate Extracellular Release of Membrane-Bound MUC2 Polymers

Cystic fibrosis is an autosomal recessive disease caused by an inheritance of two mutated *CFTR* alleles, a gene which codes for the CFTR chloride transport protein. When a pair of mutated alleles are present, the result is a dysfunctional CFTR channel which leads to the typical cystic fibrosis phenotype: thickened mucus secretions on mucus-associated epithelial surfaces throughout the body, including the lungs, small intestine and colon [77]. In the small intestine alone, this causes thick adherent mucus to impair absorption [17] and increases the likelihood of small intestinal bacterial overgrowth (SIBO) [78] and liver and pancreas damage caused by mucus blocking the common bile duct [79]. While CFTR is primarily a chloride transport protein, it has also been shown to be involved in bicarbonate (HCO_3_^−^) transport both directly, via the same pathway as Cl^−^ (though with less efficacy [80]), and indirectly, through Cl^−^/HCO_3_^−^ transport channels of the SLC26 family [81] with intestinal SLC26 HCO_3_^−^ transport dependent on a functional CFTR protein [82]. Quinton (2008) [83] hypothesised that the link between the dysfunctional CFTR channel and the cystic fibrosis phenotype in the small intestine is the presence of, or lack of, HCO_3_^−^. 

Gustafsson et al. [17], in a 2012 study, investigated the hypothesised link between bicarbonate, CFTR, and the CF phenotype in the small intestine. Mucus properties of ileal tissue explants from wild type (WT) mice and those with the CF phenotype (induced by cftrΔF508) were analysed using an Ussing-like horizontal perfusion chamber. They found that the ileal mucus layers of WT and CF mice differed significantly. WT mice displayed a thin, loose, easily aspirated mucus, whereas the mucus of CF mice was thick, firmly attached, and not easily removable. The WT mucus was also far more permeable to 2 µm diameter beads, which penetrated deep into the mucus layer, reaching as far as the crypt openings. Beads applied to the CF mucus barely penetrated any further than the villus tip. To initially investigate the bicarbonate hypothesis, bicarbonate was added to apical buffers of CF ileal explants in concentrations of 23, 69, 92 and 115 mM. It was found that the mucus layer remained attached at 69 mM, but at 92 and 115 mM it could be aspirated, with percent aspiration increasing from 50% at 92 mM to 75% at 115 mM. The result achieved when using 115 mM bicarbonate was not significantly different to what was observed in WT mice, indicating that the presence of bicarbonate restores the CF mucus phenotype to the WT. Gustafsson et al. continued the investigation by assessing the modulation of the mucus layer by secreted bicarbonate—of which the lack of, due to a dysfunctional CFTR protein, forms the backbone of the bicarbonate/CF hypothesis. To do this, bicarbonate was removed from the serosal side of WT ileal tissue explants. This resulted in a mucus layer that could not be aspirated from the epithelia, indicating that secreted bicarbonates are important for mucus layer function. A similar experiment on CF explants was done by instead adding 115 mM of bicarbonate to the serosal side, which did not cause a change in the mucus layer, suggesting that CFTR is an important regulator of basolateral to apical bicarbonate transport. After theorising that a potential action of bicarbonate ions is to chelate the Ca^2+^ ions which form cross-links within the mucin polymers, allowing the mucins to unfold, a final experiment was done using ethylenediaminetetraacetic acid (EDTA), a well-known chelator of Ca^2+^ ions [84]. EDTA was added to the apical side of the ileal explant of CF mice in concentrations of 0, 10 and 20 mM. Here it was found that 20 mM of EDTA rescued the CF phenotype mucus layer to the same degreeas the application of 115 mM bicarbonate, indicating that a role of bicarbonate is indeed as a chelator of mucin-bound calcium ions. Overall, Gustafsson et al. have given compelling evidence for the role of secreted bicarbonates in the formation of a functional small intestinal mucus layer, as well as suggesting a role of bicarbonate in mucus layer formation through calcium chelation. The chelation of Ca^2+^ ions, however, which allows mucus to unfold and hydrate, did not fully explain why the mucus could not be removed from the epithelia.

Schütte et al. (2014) [85] theorised that the missing link was in fact protease activity by meprin β, a type 1 transmembrane zinc endopeptidase (metalloprotease) from the astacin family [86], and known to be expressed throughout the small intestine [87]. They hypothesised that meprin β works in combination with CFTR-associated bicarbonate secretions to release membrane-bound MUC2 polymers from the epithelium. At this stage, the idea that MUC2 is secreted in a membrane-bound form was an exciting new concept that had not been explored.

Previous work from Lottaz et al. (1999) [87] has identified the structure of the meprin metalloprotease in the small intestine and colon using immunostaining and in situ hybridisation techniques. Using subunit-specific mRNA and antibody probes, they found that two subunits, α and β, were expressed in the epithelia of the small intestine, while the β subunit was not expressed in the colon. The two subunits have been identified previously, and noted to be expressed in zymogen form, requiring activation prior to carrying out their action via the proteolytic removal of an activation peptide [88]. Antibody staining suggested that small intestinal meprin subunits form membrane-bound dimers with the β subunit associated with the membrane. The expression of the meprin dimer was also limited to villi cells despite the presence of mRNA within the crypt cells, with staining indicating presence in the brush border of enterocytes, the most dominant cell of the small intestinal epithelium. They also noted that the full expression of the protein seems to begin at the end of the crypt at the crypt/villus junction. The lack of meprin β in the colon led to α subunit translocation towards the intestinal lumen, where in the small intestine the α subunit remained closely associated with the epithelia. The physiological differences in meprin expression between the small intestine and colon suggest that this metalloprotease may have a regulatory role in the processes of both organs. One role for meprin in the colon is explored by Banerjee et al. (2011) [89] and Banerjee and Bond (2008) [75], where meprin β deficient mice were protected against dextran sulphate sodium (DSS)-induced colitis by preventing the activation of IL-18, an inflammatory cytokine, from its inactive form. The absence of meprin α also aggravated DSS-induced colitis, indicating an inflammatory role for meprin α in the colon. This may explain the absence of meprin β expression in the colon and increased shedding of meprin α into the lumen, thereby optimising the inflammatory response.

With the work of Gustafson et al. [17] and Lottaz et al. [87] as the support for their study, Schütte et al. (2014) [85] studied the effects of a dual-allele meprin β knockout (Meprin β^−/−^) on the small intestinal mucus layer of mice, and compared findings against the phenotypes of WT and CF (cftrΔF508) mice. Mice ileal tissue explants were mounted on Ussing chambers for analysis. To follow on from their hypothesis, if meprin β is suggested to cleave mucins from the membrane under the presence of CFTR-associated bicarbonate ions, then, other than the wild type, there should be two distinct phenotypes (Figure 2).

Meprin β^−\−^:

Mucus remains anchored (not able to be aspirated) as the metalloprotease is not present. The presence, however, of a functional CFTR and the associated bicarbonate secretions should allow mucus expansion somewhat due to the chelation of calcium, given the evidence shown in the paper by Gustafsson et al. The expansion of anchored mucus in this way will likely lead to an increase in permeability. The addition of bicarbonate should not normalise the mucus layer, as seen in the Gustafsson et al. paper. The addition of recombinant meprin β should normalise the mucus layer by releasing the mucus from the membrane.

CFTR^−\−^:

The absence of a functional CFTR should not allow mucus expansion nor cleavage, and the presence of calcium within the mucus will keep the mucus in a packed, dense state preventing the permeation of larger bacteria-sized particles. Adding bicarbonate should normalise the mucus layer by allowing unfolding, but adding a meprin β inhibitor should prevent any release from the membrane by inhibiting the anchor point cleavage by meprin β.

What was found by Schütte et al. satisfied the above hypotheses. The absence of meprin β in the meprin β^−/−^ mouse ileal tissue caused the development of a 200 µM thick mucus layer that could not be aspirated, similarly to that of CF mice, and significantly different to that of the WT, of which around 85% of mucus could be aspirated. The meprin β^−/−^ mucus was also more penetrable to fluorescent 0.5 to 1 µm beads than that of the CF mice, suggesting that the meprin β^−/−^ was expanded, therefore being more permeable than CF mucus, but remained attached to the epithelium. This fits with the hypothesis that the absence of meprin β in mouse ileal tissue should give a mucus layer that is expanded but not removable as the MUC2 mucin polymers cannot be cleaved from the membrane. Following the aspiration of mucus from the WT, a known meprin β inhibitor, actinonin [90], was added to the epithelia, which resulted in newly formed mucus to be attached and not removable. Moreover, bicarbonate addition to the CF mucus layer caused normalisation, but the co-administration of bicarbonate with actinonin prevented the detachment of CF mucus, and the addition of recombinant meprin β caused the detachment of meprin β^−/−^ mucus; all results further contribute to the idea that meprin β activity does indeed cleave mucus from the membrane. In this study, Schütte et al. also expressed three recombinant fragments of MUC2 mucins to investigate which areas of the MUC2 protein backbone are susceptible to meprin β-mediated cleavage. These fragments were the MUC2 N-terminus, consisting of a signal peptide sequence, vWF D1, D2, D’ and D3, a short PTS domain and CysD1; the CysD2 domain; and the C-terminal globular structure. Exposing these fragments to meprin β revealed that the C-terminus and CysD2 fragments were unaffected, but the MUC2 N-terminal fragment was cleaved, giving bands of 130 kDa and 110 kDa upon SDS-PAGE. The undigested MUC2 N-terminus gave a band above the 250 kDa ladder marker, indicating that this protein was larger than 250 kDa in size. The 130 kDa and 110 kDa were subjected to EDMAN sequencing to determine the amino acid sequence of these fragments. The EDMAN sequencing of meprin β MUC2 N-terminal digests suggests that the enzyme cleaves at _686_SHCLE_690_ and _754_LIGQS_758_ in the TIL2 and E2 of the vWF-like D2 domain, respectively. The cleavage of the protein backbone at _686_SHCLE_690_ agreed with previous data on meprin β cleavage sites, but _764_LIGQS_758_ did not [91]. Based on these cleavage sites and the MUC2 amino acid structure published by Svensson et al. (2018), a protein fragment of 65 amino acids starting with G691 and ending with S759 should be produced, giving a molecular weight of approximately 7000 Da. The identification and sequencing of this fragment should be done to clarify whether meprin β indeed cleaves the MUC2 D2 domain at _686_SHCLE_690_ and _754_LIGQS_758_.

The identification of the cleavage of the MUC2 N-terminal structure in this way prompted the hypothesis that MUC2 mucins are secreted bound to the membrane at the N terminal side of mucin polymers by an unknown mechanism, and that the presence of bicarbonate allows the cleavage site in the D2 domain to unfold, giving meprin β access. The resulting activity is suggested to be the release of MUC2 mucin polymers from the membrane by cleavage at sites within the D2 domain, allowing the trimers or dimers formed at the D3 domain to remain intact and progress to form the extracellular mucus layer depicts three separate scenarios of the post-secretory organisation of MUC2 mucin polymers based on the research of meprin β and CFTR in MUC2 mucin post-secretory processing. The first is a WT scenario, in which CFTR provides the bicarbonate necessary to unfold the meprin β cleavage site and allow MUC2 to be cleaved from the membrane; the second is a CF scenario, wherein a dysfunctional CFTR channel does not provide the bicarbonate necessary for whole mucin polymer expansion including the meprin β cleavage site, resulting in a thick, adherent mucus with low permeability; and the third scenario is a meprin β knockout, where the mucus layer unfolds due to a functional CFTR channel providing bicarbonate, but a lack of meprin leaves the mucus anchored to the epithelial tissue. These diagrams have been produced considering the consensus method of MUC2 polymerisation described in the previous section, which includes the formation of five or six-sided ring-like structures.

Interestingly, results from germ-free mice give an indication that the presence of the small intestinal microbiome is important for the activity of meprin β. Germ-free mice were found to have a phenotype similar to that of meprin β^−/−^, but immune staining showed that meprin β was abundant, though in a much closer proximity to the epithelial tissue. Immunoblots of cell lysates showed that germ-free mice contained more membrane-bound meprin β (140 kDa), as opposed to the cleaved and shed meprin β (around 85 kDa). After the colonisation of the small intestine of germ-free mice with the microbiome of conventionally raised mice, the mucus layer normalised within 6 weeks. This suggests that meprin β must be cleaved and shed prior to activity, and that the presence of the microbiome mediates this cleavage. Schutte et al. suggested that this may be due to disintegrin and metalloprotease ADAM10/17 activity, as it has been shown previously that ADAM10/ADAM17 release meprin β from the cell surface [92]. However, they found no difference in mRNA levels of ADAM10 or ADAM17 between germ-free and WT mice, respectively, suggesting that these proteins may not have as large an impact as previously thought. Though mRNA levels may not be the key indicator of activity, bacterial components, such as lipoteichoic acid (LTA) from the cell wall of gram-positive bacteria like *Staphylococcus aureus*, are known to activate ADAM10 through the activation of platelet-activating factor receptor [93].

ADAM proteins are membrane-anchored disintegrin metalloproteases which, when in catalytically active form, function as ectodomain sheddases that cleave transmembrane proteins in the Type I or Type II group, with some additional specificity for glycosylphosphatidylinositol (GPI)-anchored proteins [94]. In the gastrointestinal tract, ADAM10 regulates intestinal development and maintenance, as well as contributing to intestinal stress responses through sheddase activity in Notch and EGFR/ErbB signalling pathways, to name a few [94]. The role of ADAM10 and 17 in the ectodomain shedding of meprin β has been explored previously [92,95], confirming that ADAM10 is a major contributor to the epithelial shedding of meprin β. The results from Schutte et al. suggested that meprin β must be shed from the epithelial surface to carry out its proteolytic functions, which is interesting considering the proposed function at the epithelial surface, not in the lumen. Wichert at al. (2017) [96] contributed further, using HEK293 cells transfected with meprin β and ADAM10/17 or matriptase-2 (MT-2), a protease which activates meprin β while in its transmembrane region by cleaving at arginine 61 [97]. A Western blot analysis of cell supernatant and lysates revealed that when MT-2 is co-transfected with transmembrane meprin β, there was no detectable presence of meprin β in the supernatant, where small amounts were observed in cells with meprin β alone (likely due to endogenous ADAM activity) and large amounts in cells transfected with meprin β and ADAM 10/ADAM 17. A co-transfection of meprin β, MT-2, and ADAM10/ADAM17 may have presented a clearer picture here, as MT-2 should activate meprin β and prevent shedding by ADAM10/ADAM17 if ADAM10/ADAM17 can only cause meprin β to be shed if it is in an inactive state. Nevertheless, the result suggests that meprin β cannot be shed by ADAM proteins following transmembrane activation, implying that it must be shed in an inactive form and activated while solubilised. Activation and shedding was suggested to be dependent on a Motif N-terminal of the EGF-like domain of meprin β, as chimeric meprin β, which contained theproposed human ADAM17 binding site was exchanged for that of murine meprin β and resulted in an impaired binding of ADAM10 and MT-2. Wichert et al. also suggest that the activity of meprin β on MUC2 is regulated by the presence of the microbiome, and provide an example of the regulation of *P. gingivalis*, which causes a dysfunctional mucus layer when it reaches the small intestine, showing that RgpB, an enzyme secreted by the pathogen, activates membrane-bound meprin β and prevents shedding, in turn preventing the cleavage of MUC2 (Figure 3).

### 5.2. Small Intestinal Mucus Layer Organisation

While the mucus layer of the colon is widely accepted as being composed of two distinct layers—a firm inner layer which is impenetrable to bacteria, and a loose sloppy outer layer—the mucus layer organisation of the small intestine is split into two schools of thought—the presence of a single loose layer of mucus, or the presence of an inner and outer layer. A higher proportion of papers, both experimental and review, support the idea of a single loose layer [17,43,85]. A study by Atuma et al. (2001) [98] provides the major support for the presence of an inner mucus layer in the small intestine. In this study, the stomach, small intestinal and colonic mucosa of anaesthetised Wistar rats were exteriorised and placed onto an illuminated, truncated Lucite cone, with the mucosa facing up. A Lucite chamber was placed over the mucosal tissue to expose a particular area, specifically either 1.2 cm^2^ or 0.9 cm^2^, respective to stomach and intestinal tissues. The chamber was filled with a physiological saline solution to keep the tissue and mucus layer moist. Analysis was done by inserting siliconized micropipettes (1–2 µm diameter), controlled by a micromanipulator, at a 35-degree angle and measuring the probes’ distance from the epithelia with a digimatic indicator. Region-specific analysis was carried out on each organ except for the colon; these included the antrum and corpus (stomach), and the duodenum, jejunum and ileum (small intestine). Colon mucosa was analysed 1–2 cm from the caecum. In the case of two distinct mucus layers being observed, inner layers were described as “firmly adherent” and the outer layers as “loosely adherent”. The firmly adherent mucus layers were categorised as not easily removed by suction. This is not to be confused with descriptions of firm inner mucus, which is often described as having different characteristics to the outer layer [44], as here the “firm adherent” does not directly implicate different properties between the two gels. The outer layer is also described as adherent, but adherent to what exactly is unclear, as only one layer is in contact with the mucosa. Clearly there is need for standardised nomenclature and definitions throughout this field of study.

Micropipette analysis revealed that a firmly adherent inner layer was present and easily identifiable in both the stomach and the colon. The antrum of the stomach had the thickest inner layer, which reached up to 154 µm in thickness and made up 56% of the total mucus thickness. The colonic inner layer was slightly smaller in thickness at 116 µm, but only made up 13% of the total mucus thickness. In the duodenum, jejunum and ileum, the thickness of the whole mucus layer varied (duodenum: 170 µm; jejunum: 123 µm; and ileum: 480 µm). Upon the removal of the loosely adherent mucus, a firmly adherent, discontinuous mucus layer that followed the grooves of the villi was identified. This mucus was described as opaque, as opposed to previously removed translucent mucus, and varied from 16 µm to 29 µm, being thickest in the ileum.

While it may be that these results point towards there being a firm layer, if the meprin β/CFTR-associated model of MUC2 post-secretory organisation is applied here, the observations in this study may be explained. The opaque mucus that is observed upon the removal of the outer layer may be MUC2 mucin polymers that have not yet undergone cleavage via meprin β or yet unfolded under the action of bicarbonate. The study shows that following the aspiration of mucus throughout the small intestine, the observed thin layer of opaque mucus can be aspirated after 90 min, leaving another layer of thin, opaque, adherent mucus. It is possible that this event is the continuous cycle of secretion, unfolding, cleavage and expansion. The inner mucus layer being discontinuous supports this, as if a firm layer was present from a functional perspective, like in the stomach and the colon, it should be a continuous layer of mucus, rather than discontinuous. An explanation for this discontinuous mucus may come from the fact that there is no meprin β expressed in the crypts [87], which would cause crypt-associated goblet cells to secrete adherent mucus until the cells migrate towards the crypt villi junction where mucin polymers can be cleaved. While it could be argued that this is what results in a distinct firm mucus presence, Strong et al. (1993) [99] noted a high level of CFTR expression in the intestinal crypts, which likely means adherent mucus could unfold somewhat without being removed from the epithelium. This lack of meprin may be due to cell maturation, as cells may not have matured to the full extent; it may also be an evolutionary adaptation which causes mucus to be ever present in the crypts, protecting the delicate stem and Paneth cells.

## 6. Discussion

This review has focused on identifying and discussing literature regarding MUC2 mucin polymerisation, secretion and post-secretory organisation. The aim of this review was to demonstrate that MUC2 mucin polymerisation, secretion and post-secretory organisation is not as well understood as previously thought. This stems from several opposing perspectives in the literature regarding these aspects, evidence from newer publications suggesting overhauls of formerly understood models of MUC2 function and novel research revealing previously unknown processes.

The first of the processes reviewed was MUC2 mucin polymerisation and secretion. Here we identified a consensus model of MUC2 polymerisation and secretion, a model which can be observed in this review to be outdated and does not align with more recent publications. These problems range from minor problems which suggest slight changes made to the model, to major problems associated with conflicting research which suggest that parts of the model could be incorrect. Considering the minor changes, an update of the consensus model could be done by replacing the current C-terminal structure (D4-B-C-CK) with the newly identified C-terminal structure of the MUC2 mucin observed in the recent publication by Gallego et al. (2023) [20] (VWCN-D4-C1-C’-C3-C4-CK). Gallego et al. [20] also discovered an additional disulphide bridge in the VWCN domain, indicating that the MUC2 C-terminus forms dimers at both the N-terminal and C-terminal side of the C-terminus at the VWCN and CK domains. This may also be included in the consensus model, though it is currently unclear whether this additional disulphide bond would form during the polymerisation of full size MUC2 mucins, as the research by Gallego et al. [20] was done using only a truncated MUC2 structure which did not include the central PTS region or the N-terminal globular region. However, it is important to add that much of the research which forms the backbone of this model, e.g., trimer formation at the D3 domains, was carried out using truncated MUC2 mucin proteins which often lacked the central VNTR region [76] as there is difficulty working with full size mucins due to their large size, and therefore, there is an argument for the representation of this extra disulphide bridge within the model.

Where major conflict arises with the consensus model is through two publications which were published by Javitt et al. in 2019 and 2020 [6,19]. In 2019 Javitt et al. [19] published a paper in which they found that the X-ray crystal structure of MUC2 formed dimers at the D3 domain, not trimers. This study was supported in a follow up study in 2020, where an analysis of the MUC2 C-terminal structure was conducted using cryo-electron microscopy, again showing that the MUC2 C-terminal structure formed dimers and not trimers. If MUC2 mucin polymers do indeed form dimers, this would result in—as suggested by Javitt et al. (2020) [6]—linear polymers similar to those formed during MUC5B and MUC5AC polymerisation [100]. This has large implications for the consensus model, as it suggests that there is no formation of 5–6-sided ring-like structures which associate non-covalently to form a mesh-like network of rings, and more likely form linear sheets of polymers. Differences between the studies by Javitt et al. (2019; 2020) [6,19], Godl et al. (2002) [76] and Nilssen et al. (2014) [18] include the use of more modern techniques such as cryo-electron microscopy. Cryo-electron microscopy, used in the Javitt et al. (2020) [6] publication, was also recently used to elucidate the MUC2 C-terminal structure in the paper by Gallego et al. (2023) [20], and it may be that this technique provides the resolution required to elucidate the structure and multimerization of the MUC2 N-terminus. However, results from SDS-PAGE and other analysis techniques including microscopy do provide significant evidence for both the formation of dimers and trimers. Another difference is the cell line used to express the truncated MUC2 proteins. Godl et al. (2002) [76] and Nisllen et al. (2014) [18] both used Chinese hamster ovary (CHO) cells, while Javitt et al. (2019; 2020) [6,19] used human embryonic kidney cells. The differences in the multimerization may be due to the difference in cell type. N-glycosylation is known to occur in the endoplasmic reticulum and supports the formation of dimers in the CK domain before translocation to the Golgi, where the D3 multimerization is expected to occur [101]. It is possible that N-glycosylation is important for the multimerization of the D3 domain, given that there are six sites for N-glycosylation in the MUC2 N-terminus, with one being in the vWF D3 domain and the other five being in the TIL-3 and E-3 domains [19]. Previous studies have noted differences in protein glycosylation by CHO cells and HEK cells. Croset et al. (2012) studied the glycosylation of 12 proteins in CHO cells and HEK cells, and for all proteins significant differences were observed between the glycosylation patterns of the same proteins produced by CHO cells and HEK cells [102]. It has been noted previously that CHO cells only produce α2,3 sialylated N-glycans, but HEK cells produce α2,3 sialylated N-glycans and α2,6 sialylated N-glycans. Moreover, humans cannot attach N-glycolylneuraminic acid, whereas many non-mammalian cells like CHO cells can [103]. Bohm et al. (2015) studied the glycosylation patterns of human coagulation factor VII (HCFVII) produced by CHO cells and HEK cells, and found that many N-glycans present on HCFVII were not found on the same proteins produced by CHO cells, and had more structural variety, less terminal sialylation, and more terminal GalNAc [104]. It may be that differences in N-glycosylation between MUC2 N-termini produced by CHO cells and HEK cells influence D3 domain multimerization into dimers or trimers. A study which included comparisons of the N-terminal multimers produced by both cells would give an answer here. It must also be considered that neither of these cells are goblet cells, and therefore they are not specialised for the production of mucins. Another possible scenario is that both trimers and dimers can form at the N-terminal domain of MUC2 in vivo. How this would impact the mucus layer in vivo is unclear. The mucus layers of the small intestine and colon share the major gel-forming component in the MUC2 mucin, but the characteristics of these mucus layers are different, with the colon having a clearly defined, generally sterile, inner layer and sloppy outer layer, whereas in the small intestine the mucus is far looser, and it is unclear whether there is an inner layer or not. It may be that the multimerization of the D3 domain into either trimers or dimers has an impact on the characteristics of the produced mucus layer.

The meprin β model provides insights into how the mucus layer undergoes post-secretory processing to promote the formation of a permeable mucus layer, a process which was previously unknown. Small intestinal mucus forms a layer which is protective and permeable to small molecules, with pore size proposed to be 500 nm, up to as far as 2000 nm in size [17,105]. The formation of these pores allows particles smaller than the pores, i.e., small oligosaccharides, oligopeptides and fatty acids, to access the brush border enzymes required for end stage digestion and particle absorption. The model, which in short suggests that luminal bicarbonate released by a functional CFTR channel is essential for the unfolding and release of MUC2 polymers from the membrane by meprin β, is a novel perspective in this area, and may be crucial for further research into understanding and treating diseases such as cystic fibrosis.

The contribution of the small intestinal microflora to the formation of a functional mucus layer, and overall functionality of the organ, if valid, is a significant factor in mucosal physiology. The formation of a dysfunctional mucus layer in germ-free mice [85], as well as the experiments on the *P. gingivalis* activation of the meprin β sheddase ADAM10 [96], suggests that microbes are direct mediators of small intestinal function. It is interesting that the small intestine seemingly would not function adequately without a microflora and may suggest that the meprin β model evolved alongside the small intestinal microbiome.

The meprin β model also gives us further insight into epithelial protection by the intestinal mucus layer. Though in the literature it is accepted that the small intestinal mucus layer forms a single sloppy layer of mucus, some literature suggests the presence of an inner layer of mucus. When analysing the physiological structures present in the small intestine, as well as its primary absorptive function, the absence of a firm layer with reduced permeability, like that seen in the colon, seems logical. Between the colon and the small intestine, there are some major physiological differences in structure and cell type. One of these major cell types of the small intestine and not the colon is the Paneth cell, found at the bottom of the small intestinal crypt [106]. Paneth cells are immune cells, secreting antimicrobial compounds into the intestinal lumen. These cells are not generally present throughout the colon despite an enormous number of microbes housed within [107]. The firm layer of the colon is devoid of bacteria [44] indicating that the inner layer is robust enough in its protective capability, indicating that Paneth cells are simply not required. If the firm layer is so efficient at protecting the epithelia, why is it not present in the small intestine? The likelihood is due to differences in the function of each organ. The small intestine is primarily an absorptive organ, and a firm layer like that seen in the colon may impede the absorption of large or complex molecules which are likely to become trapped in a denser mucus layer. It is likely that the abundance of Paneth cells throughout the small intestinal crypts creates a high concentration of antimicrobial compounds at the epithelial surface, protecting the epithelia from microbes. The localisation of these cells is also important, as they sit at the base of the crypt next to the stem cells. This localisation may be because they are essential cells for mucosal protection or are essential for the protection of stem cells. However, while there is evidence to suggest that the firm layer is impenetrable to bacteria [44], there is little information present on the pore size of the inner layer of colonic mucus. Understanding the size of these pores is critical for a further understanding of the mucus layers of both organs. Peyer’s patches in the ileum also back up the logical assumption of a single sloppy layer of mucus in the small intestine. Along the length of the small intestine, the bacterial count increases significantly, from ×10^4^ CFU/mL in the proximal jejunum, to up to ×10^9^ CFU/mL in the terminal ileum [108]. Despite the significant number of microbes present in the terminal ileum, which is not too far away from colonic levels, there is no observable, firm attached layer of mucus like that of the colon [98]. What is unique to the ileum, however, in respect to other sections of the small intestine, is an extremely high density of submucosal Peyer’s patches, regulators of the small intestinal immune response [109]. The lack of a firm layer, but the Increase in immune cells, further supports that these cells are a surrogate for the firm inner mucus layer, since it is likely that the firm mucus layer may impede the absorption process when concerning larger and more complex molecules when compared to the loose sloppy mucus layer of the small intestine. Also observed in the ileum is an increased mucus layer thickness, likely to increase the distance of the bacteria from the epithelia [98]. Ermund et al. (2013) [105], with the use of bacteria-sized beads and rat small intestinal tissue explants, showed that beads in the ileum could get close to the epithelia in explant tissue, but the presence of actual bacteria in the explants was never in direct contact. It is likely that peristalsis and mucus renewal are effective mediators of bacterial penetration, continually moving bacteria up and away from the epithelium, whilst allowing smaller particles like those for brush border metabolism to reach the epithelia. Atuma et al. (2001) [98], as previously described, noted the presence of a “firmly adherent mucus layer” beneath the sloppy mucus layer of the duodenum, jejunum and ileum. If the meprin β model is accounted for, this adherent layer may be unfolded and attached mucus which has not yet been removed from the membrane by meprin β. However, meprin β expression is absent from crypt cells, which would likely result in crypt goblet cells secreting attached mucus. While this may not necessarily be a firm layer, as the presence of CFTR within the crypt would allow the mucins to unfold, it may be an evolutionary mechanism to ensure that the crypts are filled with mucin at all times, protecting the essential stem cells at the base of the intestinal crypt.

The significance of the various findings discussed in this review should not be underestimated, as these identify that the formation of a mucus layer in the small intestine is much more complex than previously thought, and also paves the way for the research of gut evolution and treatments for intestinal disease. Nevertheless, there are some areas of concern, and the requirement for further supporting research in this field is critical for complete mucus layer characterisation. Regarding further research, it remains unclear the mechanisms involved which link the MUC2 polymers to the membrane post-secretion, and this link is key to the completion and validation of the meprin β model of MUC2 secretion. Moreover, there are some areas associated with the meprin β mediation of MUC2 membrane cleavage which can be met with concern. Firstly, Schutte et al. [85] demonstrate with microscopy that MUC2 seemingly is anchored to the goblet cells. However, both Gustafsson et al. [17] and Schutte et al. use Carnoy-fixation techniques to observe the small intestinal epithelia and mucus using microscopy. While the data look appealing, specifically images of MUC2 anchored to goblet cells, fixing tissues using this technique significantly dehydrates tissues which may influence a conformational change in the mucus layer [110], and could affect the validity of the work. Also, it is unclear how the MUC2 mucin polymers are anchored to the membrane, and since MUC2 mucin monomers all contain the same protein structure, how do the goblet cells determine which parts of the mucin polymer are anchored? Are there multiple anchors per polymer, or is there a single anchor per polymer? These are all questions which require more research into this area.

The use of gene knockout and the expression of recombinant proteins using expression systems can be unreliable in producing a correct and functional product. Godl et al. (2002) [76] noted that CHO cells have some difficulty expressing the recombinant N-terminal MUC2 protein. A similar recombinant protein was used in Schutte et al. (2014) [85] to investigate the meprin β cleavage site in MUC2. An incorrect folding of the recombinant MUC2 N-terminus may have resulted in the attained result—a result which indicated two cleavage sites, one of which was not supported by previous literature. Moreover, the source of the recombinant meprin β in the same study is not clarified, and thus cannot be subject to critique. As the recombinant MUC2 protein was produced with difficulty in CHO cells, if we assume that meprin β was produced using the same method, it is not farfetched to assume that the metalloprotease could be similarly impaired. Knockout techniques can be subject to a similar critique. Utilising homologous recombination as a method of knockout is associated with random, non-specific genomic vector insertion, potentially disrupting other genes involved [111,112]. Moreover, in bead permeability studies in meprin β^−/−^ mice, the permeability of the meprin β^−/−^ mucus layer is closer to the WT than that seen in CF mice. How much closer the permeability is to the WT is not clear. On observation, the WT mice have clear fluorescent particles interspersed throughout the mucus layer, and the CF mice, as expected, have very few particles within, with most sitting atop the mucus layer. Meprin β^−/−^ mucus is comparable to both, but one may argue that significantly more beads sit atop the mucus layer than interspersed within. This indicates that the knockout of meprin β has also caused an impairment of mucus unfolding towards the luminal side, not just at the epithelial cleavage point. It could be hypothesised that meprin β also functions to organise the mucus layer further to allow a higher permeability, though the mechanisms for this are unclear. As the knockout of meprin β has also caused an impairment of mucus unfolding towards the luminal side, not just at the epithelial cleavage point, this may indicate that meprin β contributes to more than just epithelial cleavage, either through proteolytic activity itself or through another activation cascade. If the metalloprotease is involved in other biological processes in the small intestine, the results seen in the knockout must be treated with caution.

Regarding the influence of bacteria on the organisation of the mucus layer, the results are promising, but do require more research. Studies on germ-free animals, which indicated that microbes mediate meprin β cleavage, were carried out on mice. The mouse and human small intestinal microbiome are very different [113], and thus, more data are required on the influence of human small intestinal microflora. Moreover, the data which contributed to the meprin β shedding on the behalf of ADAM proteins was gained through the study of pathogenic organisms, specifically *P. gingivalis* [96] and *S. aureus* [93]. The influence on the small intestinal mucus layer by commensal and symbiotic microbes in the human gastrointestinal tract should be the focus of further research, as pathogenic organisms are likely to have mucus disruption mechanisms as a method of infection, rather than physiological processes of the small intestine reacting to these microbes to promote mucus layer formation. The way the microbiome contributes to the function of the mucus layer is key to understanding the relationship between the small intestine and the luminal microbes. If the presence of bacterial cells is the contributing factor, their presence close to the epithelia would be detrimental to the epithelia through the risk of infection.

## Figures and Tables

**Figure 2 polymers-16-01663-f002:**
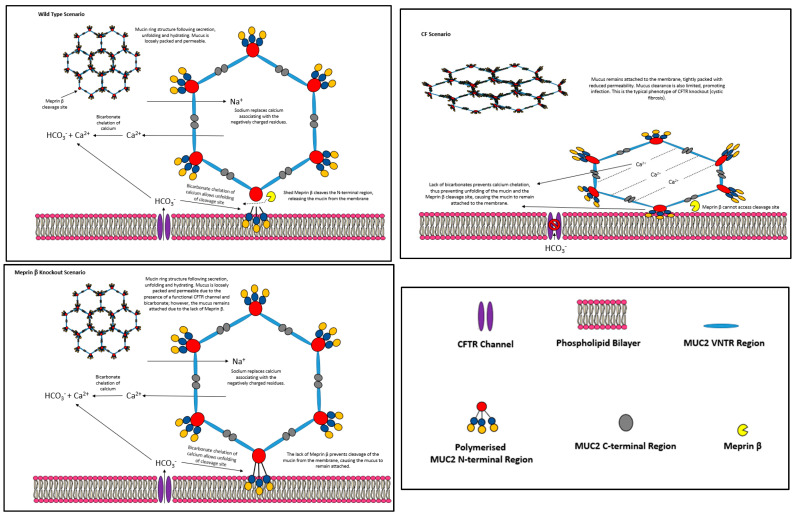
Figure depicts the post-secretional processes involved in cleaving the MUC2 mucin from the membrane based on the Schutte et al. [85] paper, specifically the wild-type scenario (above) and the CF scenario (below). *Wild Type Scenario:* The MUC2 mucin is secreted attached to the membrane at the N-terminus. CFTR-associated bicarbonate chelates mucin cross-linking Ca^2+^ ions, which are replaced with sodium ions. Bicarbonate allows unfolding of the meprin β cleavage site, and shed meprin β cleaves the mucin polymer from the membrane before to the D3 domain, leaving behind the D1 and D2 domains. CF Scenario: A lack of bicarbonate prevents chelation of cross-links and thus prevents mucin unfolding. Meprin β cleavage site remains compact, preventing cleavage and leaving the mucins attached to the epithelia in a compact form. Meprin β Knockout: mucin polymers expand due to the presence of a functional CFTR channel which provides the bicarbonate required for expansion. However, the absence of meprin β causes the mucin polymers to remain attached to the epithelial tissue.

**Figure 3 polymers-16-01663-f003:**
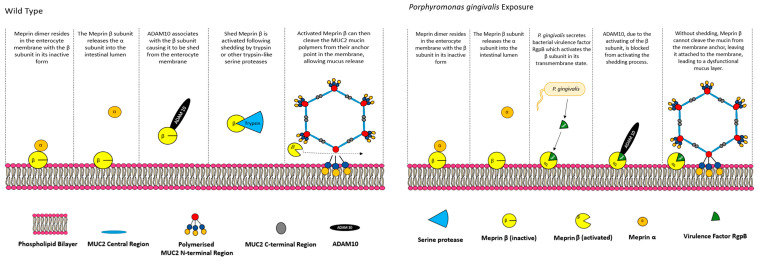
The meprin β model of cleavage in wild type and disease scenarios based on the work by Schutte et al. (2014) [85] and Wichert et al. (2017) [96]. (Wild Type) Meprin β resides within the enterocyte membrane as a dimer with the α subunit. The α subunit is released into the intestinal lumen, and ADAM10 associates with the membrane-bound meprin β subunit to release it into the intestinal lumen. A tryptic protease activates meprin β which goes on to cleave MUC2 from the intestinal epithelium. (*P. gingivalis* Exposure) Meprin β resides within the enterocyte membrane as a dimer with the α subunit. The α subunit is released into the intestinal lumen. *P. gingivalis* present in the intestine produces virulence factor RgpB which binds to meprin β, preventing its release from the membrane by ADAM10. The lack of released meprin β causes MUC2 polymers to remain anchored to the intestinal epithelium due to the lack of meprin β-mediated cleavage. The MUC2 polymer structure here is based on the consensus model put forward by the literature [7].

**Table 1 polymers-16-01663-t001:** Membrane-tethered cell surface, secreted gel-forming, and secreted non-gel-forming mucins.

Membrane-Tethered Cell Surface	Secreted Gel-Forming	Secreted Non-Gel-Forming
MUC1	MUC2	MUC7
MUC3A	MUC5AC	MUC8
MUC3B	MUC5B	MUC9
MUC4	MUC6	
MUC12	MUC19	
MUC13		
MUC14		
MUC15		
MUC16		
MUC17		
MUC18		
MUC19		
MUC20		
MUC21		
MUC22		

Data were collected from Pearson et al. (2016) [7] and Corfield (2015) [8].

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
