# Peer review of "The MUC2 Gene Product: Polymerisation and Post-Secretory Organisation—Current Models"

_polymers, 2024, doi:10.3390/polym16121663_

Round 1

Reviewer 1 Report

Comments and Suggestions for Authors

The authors has presented a comprehensive overview of Muc2 mucins, in particular with emphasis on integrating recent detailed structural characterization in the perspective of previous consensus model for intermolecular association (trimer), and thus, allude to the difference arising for the dimer association model. The authors outline the basis for the alternative model in a balanced way that also should stimulate further investigations. Clearly being at the crossroads between biology and polymer science, readers mainly interested in the polymer dimension may find the presentation a bit biased towards the details of the biological word. If possible, a more clear communication of the polymer aspect is recommended to be better aligned with the scope of the journal.

Comments on the Quality of English Language

The English is just fine, but a small number of misprints were spotted: "The" in the first sentence of the introduction should be omitted. The organelle Golgi is also spelled "golgi"; Not all citations appear complete/not correct style (e.g. numbers 21, 32, 41, 50, 87, 107, 115) 

Reviewer 2 Report

Comments and Suggestions for Authors

The manuscript titled " The MUC2 Gene Product: Polymerisation and Post-Secretory Organisation – Current Models" submitted as a review one, worths the publication in the Polymers MDPI journal. 

While the manuscript covers broad aspects of mucin biology, its primary contribution lies in the detailed and updated discussion of MUC2. By providing a focused analysis of MUC2 polymerization and post-secretory organization, the manuscript offers valuable insights and updates to the field. 

To enhance readability and maintaining the balance of the manuscript, the introductory sections: ”mucin structure” and “membrane-tethered mucins” should remain succinct, I mean to reduce them (or move some part in the introduction section). In this way, the main focus, or the main goal, will remain the in-depth examination of MUC2, as the manuscript title suggests as well.
